# Randomized, double-blind, placebo-controlled trial of rapamycin in amyotrophic lateral sclerosis

Jessica Mandrioli [1,2,17] ✉, Roberto D'Amico[3,4,18], Elisabetta Zucchi[2,5,17], Sara De Biasi[4,17], Federico Banchelli[3,17], Ilaria Martinelli[2,6], Cecilia Simonini[2], Domenico Lo Tartaro[4], Roberto Vicini [3], Nicola Fini[2], Giulia Gianferrari [1,2], Marcello Pinti[7], Christian Lunetta[8,9], Francesca Gerardi[8], Claudia Tarlarini[8], Letizia Mazzini[10], Fabiola De Marchi [10], Ada Scognamiglio[10], Gianni Sorarù[11,12], Andrea Fortuna[11], Giuseppe Lauria[13], Eleonora Dalla Bella[13], Claudia Caponnetto[14], Giuseppe Meo[14], Adriano Chio[15], Andrea Calvo [15] & Andrea Cossarizza [4,16,18]

In preclinical studies rapamycin was found to target neuroinflammation, by expanding regulatory T cells, and affecting autophagy, two pillars of amyotrophic lateral sclerosis (ALS) pathogenesis. Herein we report a multicenter, randomized, double-blind trial, in 63 ALS patients who were randomly assigned in a 1:1:1 ratio to receive rapamycin 2 mg/m²/day,1 mg/m²/day or placebo (EUDRACT 2016-002399-28; NCT03359538). The primary outcome, the number of patients exhibiting an increase >30% in regulatory T cells from baseline to treatment end, was not attained. Secondary outcomes were changes from baseline of T, B, NK cell subpopulations, inflammasome mRNA expression and activation status, S6-ribosomal protein phosphorylation, neurofilaments; clinical outcome measures of disease progression; survival; safety and quality of life. Of the secondary outcomes, rapamycin decreased mRNA relative expression of the pro-inflammatory cytokine IL-18, reduced plasmatic IL-18 protein, and increased the percentage of classical monocytes and memory switched B cells, although no corrections were applied for multiple tests. In conclusion, we show that rapamycin treatment is well tolerated and provides reassuring safety findings in ALS patients, but further trials are necessary to understand the biological and clinical effects of this drug in ALS.

With a life-long risk of 1:400, Amyotrophic Lateral Sclerosis (ALS) is the 3rd most common neurodegenerative disease, holding an estimated increase of 69% in the upcoming years[1]. No specific pharmacological treatments modify the disease relentless course leading to death within 3–5 years from symptoms onset. The only approved drug in Europe, riluzole, extends survival for a few months at best[2]. Despite extreme heterogeneity and intricate pathobiology, protein misfolding

and immune system dysfunction are two acknowledged pillars of ALS pathogenesis, which present in the early stages in all patients and constantly evolve to the terminal phases, therefore representing promising therapeutic targets[3].

Rapamycin, a drug used to prevent renal transplantation rejection, inhibits mammalian Target of Rapamycin Complex 1, leading to regulatory T lymphocytes (Treg) expansion and autophagy

enhancement. In fact, mTOR inhibits the induction of Tregs, a specific cell population that down-regulates immune system activation and is found to be reduced and dysfunctional in ALS patients[3]. As a consequence, self-sustaining inflammatory cytokines are upregulated and peripheral immune cell migration into the brain is promoted[4]. In ALS patients, percentage of Tregs in the blood inversely correlated with progression rate[5], and FoxP3 levels were early predictors of ALS progression and survival[6]. Thus, Tregs may be considered important therapeutic targets in ALS addressed by rapamycin.

Furthermore, several in vitro and in vivo studies also suggest that inhibition of mTOR by rapamycin may target autophagy, which is crucial not only for cell-autonomous clearance mechanisms, but also for limiting detrimental and uncontrolled activation of inflammasomes[7]. In ALS, accumulation of aggregates drives caspase-1-mediated proteolytic cleavage and secretion of proinflammatory cytokines, that further amplify inflammatory responses, resulting in chronic inflammation, tissue damage and cell death[8]. Rapamycin enhances the autophagic degradation of various aggregate-prone proteins with subsequent reduction of their toxicity in cellular or animal models not only in ALS, but also in other neurodegenerative diseases e.g., Huntington disease[9–11]. Autophagy enhancement by rapamycin is mediated by the unc-51-like kinase 1 complex and the formation of autophagosome from the phagophore[10].

In preclinical ALS studies, rapamycin reduced TDP-43 fragments accumulation and restored TDP-43 nuclear localization in cell lines[12] and in human stem cell-derived neurons and astrocytes with mutant TDP43, where rapamycin enhanced survival[9]. A microfluidic approach using rapamycin rescued the ALS motor neuron phenotype in 2D and 3D environments from a TDP-43 mouse[13]. Early rapamycin administration determined phenotype amelioration in TDP-43 mouse[14], *SQSTM1* knock-down zebrafish[15] and in Drosophila with *VAPB* mutation[16] (details in supplementary appendix, Section 1).

Here we report the results of RAP-ALS, a phase 2, multicenter, randomized, double-blind, placebo-controlled clinical trial (RCT), that evaluates safety, biological and clinical effects of oral rapamycin in persons with ALS[17].

## Results
### Participants and follow-up
From 05/10/2017 to 02/01/2020 a total of 70 patients with ALS were screened for eligibility, of whom 63 were randomly assigned to a trial group: 21 to rapamycin 2 mg/m²/day, 21 to rapamycin 1 mg/m²/day and 21 to placebo (Fig. 1). Seven patients dropped out from the study during the treatment period, whereas 15 did not conclude the follow up (after treatment). Two patients did not take at least 80% of the study drug as planned per protocol. As these two patients dropped out between week 18 and 30, Per Protocol (PP) and Intention To Treat (ITT) analyses differ at week 8 and 18 only, whereas the results of the two analyses were identical at week 30 and 54.

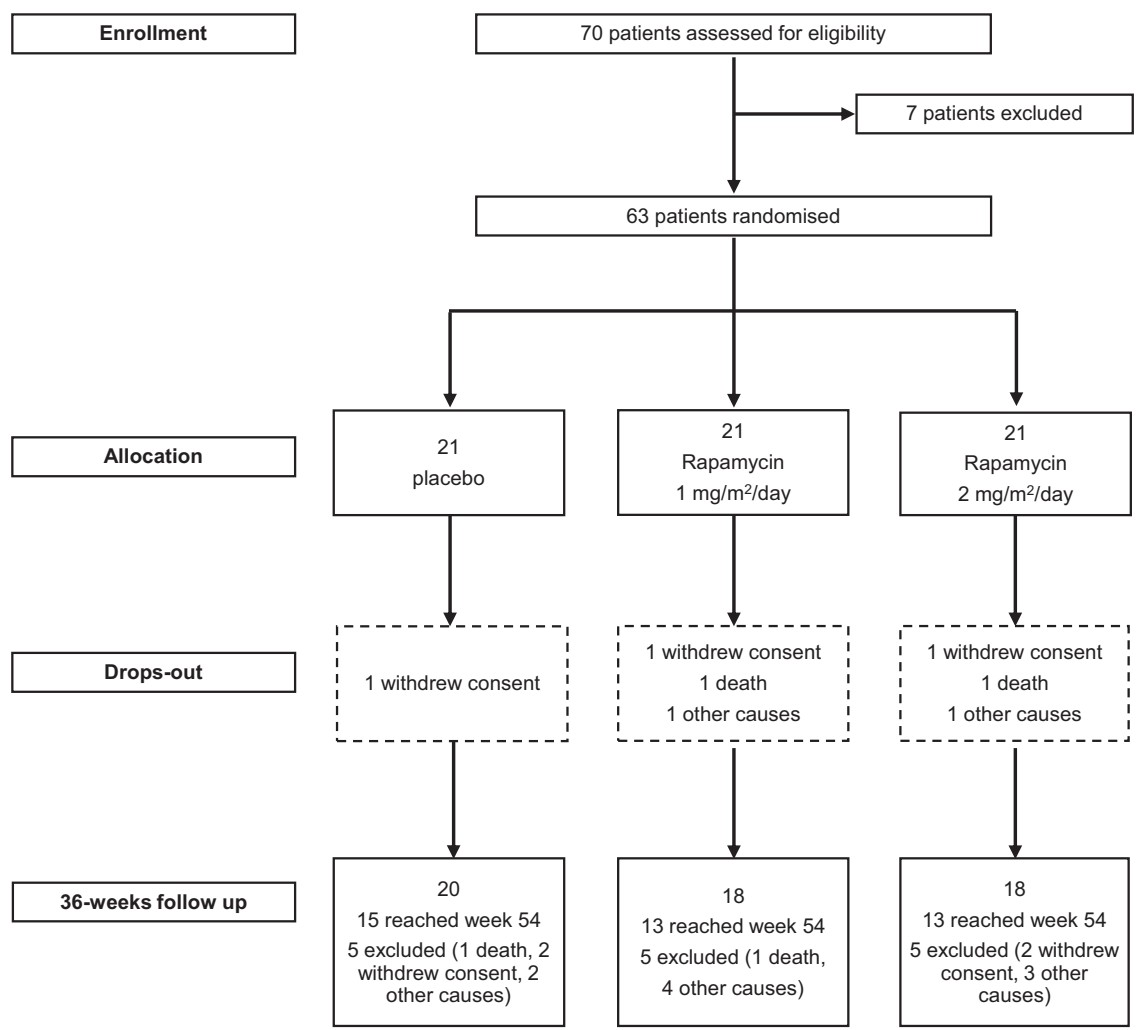

**Fig. 1 | CONSORT diagram of the study reporting screening, randomization and follow-up of ALS patients enrolled in the trial.** Source data are provided as a Source Data file.

**Table 1 | Demographic and clinical characteristics of the participants at baseline (intention-to-treat population)**

| Characteristic | Rapamycin 2 mg/m²/d (N = 21) | Rapamycin 1 mg/m²/d (N = 21) | Placebo (N = 21) |
|---|---|---|---|
| Male sex – n. (%) | 11 (52.4) | 7 (33.3) | 13 (61.9) |
| Age – years | 56.04 ± 11.88 | 55.08 ± 10.99 | 55.55 ± 12.88 |
| Body-mass index, kg/m² | 25.39 ± 4.43 | 25.63 ± 4.75 | 24.71 ± 2.73 |
| BSA, m² | 1.84 ± 0.17 | 1.81 ± 0.23 | 1.83 ± 0.22 |
| Bulbar onset – n. (%) | 4 (19.0) | 4 (19.0) | 6 (28.6) |
| Upper limb onset – n. (%) | 10 (47.6) | 9 (42.9) | 8 (38.0) |
| Lower limb onset – n. (%) | 9 (42.9) | 10 (47.6) | 10 (47.6) |
| Months since ALS symptom onset | 11.90 ± 3.95 | 14.10 ± 3.66 | 11.67 ± 4.25 |
| ALSFRS-R total score | 39.05 ± 4.17 | 38.29 ± 4.36 | 38.62 ± 5.36 |
| Bulbar score | 11.00 ± 1.61 | 10.71 ± 1.68 | 10.57 ± 1.72 |
| Fine-motor score | 8.29 ± 1.98 | 8.10 ± 2.47 | 8.38 ± 2.75 |
| Gross-motor score | 8.00 ± 2.86 | 7.67 ± 2.78 | 7.90 ± 2.79 |
| Breathing score | 11.76 ± 0.70 | 11.81 ± 0.40 | 11.76 ± 0.89 |
| Prebaseline ALSFRS-R slope[a] | 0.80 ± 0.42 | 0.75 ± 0.41 | 1.05 ± 1.00 |
| Forced vital capacity — % of predicted normal value | 95.47 ± 18.27 | 94.86 ± 17.85 | 95.04 ± 15.72 |
| MRC upper-limb score — % of predicted normal value | 86.79 ± 14.22 | 85.57 ± 10.94 | 85.27 ± 12.84 |
| MRC lower-limb score — % of predicted normal value | 85.44 ± 17.02 | 85.68 ± 18.29 | 84.90 ± 17.14 |
| MRC neck score — % of predicted normal value | 97.38 ± 5.15 | 92.62 ± 14.02 | 95.95 ± 9.17 |
| ALSAQ40 total score | 32.56 ± 14.51 | 30.67 ± 14.64 | 32.98 ± 17.72 |
| Edaravone treatment – n (%) | 0 (0.0) | 2 (9.0) | 5 (23.8) |

Source data are provided as a Source Data file.
*ALSFRS-R* Amyotrophic Lateral Sclerosis Functional Rating Scale Revised, *MRC* Medical Research Council scale, *BSA* body surface area, *ALSAQ40* Amyotrophic Lateral Sclerosis Assessment Questionnaire.
[a]Prebaseline ALSFRS-R slope has been calculated as monthly decline of ALSFRS-R score assuming a total score of 48 at onset.

**Table 2 | Patients exhibiting a positive response (increase in Treg of at least 30%), comparing baseline and treatment end (week 18) between rapamycin and placebo arm. Intention to Treat analysis**

| | Positive response | | Not positive response | | Unadjusted analysis | | | Adjusted analysis[a] | | |
|---|---|---|---|---|---|---|---|---|---|---|
| | n | % | n | % | RR | CI | p | OR | CI | P |
| Placebo | 2 | 11.8 | 15 | 88.2 | - | - | - | - | - | - |
| Rapamycin 1 mg/m²/d | 5 | 27.8 | 13 | 72.2 | 2.36 | 0.42–13.12 | 0.2365 | 3.53 | 0.31–40.80 | 0.2477 |
| Rapamycin 2 mg/m²/d | 3 | 20.0 | 12 | 80.0 | 1.70 | 0.26–11.21 | 0.5220 | 2.76 | 0.20–37.58 | 0.3836 |
| Rapamycin | 8 | 25.0 | 25 | 75.8 | 2.06 | 0.49–8.65 | 0.2961 | 3.19 | 0.43–23.66 | 0.2567 |

Source data are provided as a Source Data file.
The unadjusted comparisons were carried out with a chi-square test without any correction and the adjusted comparisons with a logistic regression model. All statistical tests were two-tailed.
*RR* relative risk, *OR* odds ratio, *CI* confidence interval.
[a]Adjusted for sex, months from onset of symptoms, ALSFRS-R slope at baseline and treatment with edaravone.

Baseline demographic and disease characteristics are summarized in Table 1. Baseline biological features of the trial participants are summarized in Supplemental materials, Table S1.

**Rapamycin effect on Treg cells**
Although 56 patients reached treatment end, only 50 blood samples from 50 participants were available for cells populations analysis at week 18 due to the COVID-19 pandemic. Among patients treated with rapamycin 1 mg/m²/d, 28% of patients presented an increase of Treg cells by at least 30%, versus 12% of patients in the placebo group (Relative Risk [RR] 2.36, 97.5% confidence interval [CI] 0.42 to 13.12; $p = 0.236$) (ITT analysis, Table 2); the positive response was observed in 20% of patients treated with rapamycin 2 mg/m²/d (RR1.70, 97.5%CI 0.26 to 11.21, $p = 0.522$). PP analysis yielded similar results (Table S2).

In an adjusted analysis, after correction for sex, time from onset, ALSFRS-R slope at baseline, and edaravone treatment, based on the comparison of differences of clinical significance between groups at baseline, odds ratio (OR) of a positive response was non-significantly

increased in Rapamycin treatment arms (Table 2). At week 18, 6/17 (35%) placebo-treated patients experienced any increase in Treg cells compared with 10/18 (56%) patients treated with rapamycin 1 mg/m²/d (RR 1.57, 97.5%CI 0.66–3.77; $p = 0.2291$) and 10/15 (67%) patients treated with rapamycin 2 mg/m²/d (RR 1.89, 97.5%CI 0.81–4.38; $p = 0.0765$).

At treatment end patients treated with rapamycin 1 mg/m²/d experienced a mean increase of Treg cells by 0.41, patients treated with 2 mg/m²/d showed stable values (−0.05), whereas patients in placebo showed a decrease in Treg cells by −0.46 from baseline (mean difference [MD] 0.87, 97.5% CI −0.36 to 2.09; $p = 0.109$ in rapamycin 1 mg/m²/d group and MD 0.41, 97.5%CI, −0.88 to 1.69; $p = 0.465$ in rapamycin 2 mg/m²/d group) (Table 3). While in the placebo arm Treg monthly variation during and after treatment showed a nearly constant decline (mean −0.10, 95%CI: −0.25 to 0.05, $p = 0.182$ during treatment; mean −0.04, 95% CI: −0.09 to 0.02, $p = 0.216$ after treatment), during treatment there was a trend towards an increase in patients who were treated with the lower dose of the drug compared to those who received the placebo (MD 0.17, 97.5%CI: −0.06 to 0.41,

**Table 3 | Changes from baseline to week 8, 18, 30, and 54 in Treg cells across treatment arms. Intention to treat analysis**

| Time point | Arm | Absolute change from baseline | | | Unadjusted analysis | | | | Adjusted analysis[a] | | | |
|---|---|---|---|---|---|---|---|---|---|---|---|---|
| | | n | mean | SD | MD | CI | | p | MD | CI | | p |
| Week 8 | Placebo | 19 | −0.35 | 1.61 | - | - | - | - | - | - | - | - |
| | Rapamycin 1 mg/m²/d | 19 | −0.22 | 1.13 | 0.13 | −0.92 | 1.18 | 0.7774 | −0.13 | −1.28 | 1.01 | 0.7868 |
| | Rapamycin 2 mg/m²/d | 18 | 0.17 | 1.44 | 0.52 | −0.55 | 1.58 | 0.2661 | 0.22 | −0.92 | 1.35 | 0.6617 |
| | Rapamycin | 37 | −0.03 | 1.28 | 0.32 | −0.47 | 1.11 | 0.4232 | 0.04 | −0.81 | 0.90 | 0.9187 |
| Week 18 | Placebo | 17 | −0.46 | 1.58 | - | - | - | - | - | - | - | - |
| | Rapamycin 1 mg/m²/d | 18 | 0.41 | 1.56 | 0.87[b] | −0.36 | 2.09 | 0.1088 | 0.53 | −0.73 | 1.79 | 0.3327 |
| | Rapamycin 2 mg/m²/d | 15 | −0.05 | 1.55 | 0.41[b] | −0.88 | 1.69 | 0.4655 | 0.09 | −1.21 | 1.39 | 0.8757 |
| | Rapamycin | 33 | 0.20 | 1.55 | 0.66 | −0.28 | 1.59 | 0.1645 | 0.32 | −0.63 | 1.28 | 0.4985 |
| Week 30 | Placebo | 12 | −0.34 | 1.51 | - | - | - | - | - | - | - | - |
| | Rapamycin 1 mg/m²/d | 15 | 0.06 | 1.21 | 0.40 | −0.81 | 1.61 | 0.4455 | −0.20 | −1.60 | 1.21 | 0.7456 |
| | Rapamycin 2 mg/m²/d | 13 | 0.12 | 1.30 | 0.45 | −0.80 | 1.70 | 0.4028 | −0.17 | −1.59 | 1.26 | 0.7862 |
| | Rapamycin | 28 | 0.09 | 1.23 | 0.42 | −0.50 | 1.34 | 0.3573 | −0.18 | −1.26 | 0.90 | 0.7342 |
| Week 54 | Placebo | 9 | −0.36 | 2.02 | - | - | - | - | - | - | - | - |
| | Rapamycin 1 mg/m²/d | 11 | −0.15 | 1.26 | 0.21 | −1.46 | 1.88 | 0.7703 | −0.57 | −2.33 | 1.20 | 0.4514 |
| | Rapamycin 2 mg/m²/d | 11 | −0.62 | 1.44 | −0.26 | −1.94 | 1.41 | 0.7100 | −0.80 | −2.44 | 0.83 | 0.2521 |
| | Rapamycin | 22 | −0.39 | 1.34 | −0.03 | −1.29 | 1.23 | 0.9634 | −0.71 | −1.98 | 0.56 | 0.2634 |

Mean absolute changes from baseline to week 8, 18, 30, and 54 are showed for each treatment group and comparison were performed using linear regression models that include indicator variables for treatment arms as the independent variables.

For the comparison of Rapamycin and placebo arms, a P value of 0.05 or less was considered to indicate statistical significance and uncertainty in results was expressed with the 95% confidence interval (CI). For the comparisons between Rapamycin 1 mg/m²/d or 2 mg/m²/d arms and the placebo arm, a P value of 0.025 or less was considered to indicate statistical significance and uncertainty in results was expressed with the 97.5% CI, to account for multiple arms comparison with the Bonferroni method. CIs were calculated based on the exact t distribution. All statistical tests were two-tailed.

MD mean difference, CI confidence interval.

[a]Adjusted analyses for sex, ALSFRS-R slope at baseline, disease duration from onset to baseline and edaravone treatment.

[b]While mean Treg cells percentage at baseline was similar in the treatment group (4.66 ± 1.86) with respect to the placebo group (4.39 ± 2.13), at week 18 Treg percentage were 4.69 ± 1.75 in the treatment group and 4.22 ± 1.62 in the placebo group. Source data are provided as a Source Data file.

$p = 0.100$ in rapamycin 1 mg/m²/d arm; MD 0.08, 97.5%CI: −0.16 to 0.33, $p = 0.450$ in rapamycin 2 mg/m²/d arm), that was not statistically significant (Table S3). In a post-hoc analysis on a limited number of samples, we found that after in vitro stimulation with anti-CD3/CD28 functionality of Treg cells resulted similar before and after therapy and not different to those from age and sex-matched healthy controls (Fig. S1).

**Treatment impact on blood cell subpopulations**
We next examined the change from baseline to each time point (week 8, 18, 30, 54) of the activation and homing capabilities of different T, B, NK cell subpopulations, comparing treatment and placebo arms (Fig. 2, Table S4); no correction was applied for multiple tests (55 outcomes were examined), therefore the following results on secondary outcomes should be interpreted cautiously.

At week 18 patients treated with rapamycin 1 mg/m²/day showed a not significant trend toward a reduction of activated (CD38+, HLA-DR+) CD8 + T lymphocytes (MD −0.92, 97.5% CI −1.88 to 0.05; $p = 0.032$), and intermediate monocytes (CD14+, CD16dim monocytes, MD −13.48, 97.5% CI −28.17 to 1.22; $p = 0.038$), with respect to placebo group from baseline to treatment end; in the same frame time this group showed an increase of the percentage of memory switched B cells (defined as IgM-, IgD-, CD21+,CD24+, CD27+, CD38- B cells, MD 0.82, 97.5% CI 0.28 to 1.37; $p = 0.002$) and of classical monocytes (CD14+, CD16- monocytes, MD 17.76, 97.5% CI 0.91 to 34.60; $p = 0.019$) (Fig. 2a–d) (Table S4).Similar results were detected at week 30 (Fig. 2g–j). In a post-hoc analysis on a limited number of samples, we found that, with

respect to controls, at baseline ALS patients showed a trend towards higher percentages of Th1 CD4 + T cells that remained similar after 18 weeks (Fig. S2).

**Rapamycin effect on inflammasome**
Patients treated with rapamycin showed lower mRNA relative expression of pro-inflammatory cytokine IL-18, which is a readout of inflammasome activation (MD −0.45, 97.5%CI −1.09 to 0.18; $p = 0.101$ for rapamycin 1 mg/m²/d group and MD −0.60, 97.5%CI −1.18 to −0.01; $p = 0.022$ for rapamycin 2 mg/m²/d group), and consistently a marked reduction in plasmatic IL-18 protein (MD −107.80, 97.5%CI −187.12 to −28.48, $p = 0.002$ for rapamycin 1 mg/m²/d group and MD −103.00, 97.5%CI −183.51 to −22.48; $p = 0.004$ for rapamycin 2 mg/m²/d group) from baseline to treatment end with respect to placebo-arm patients (Fig. 2e, f, Table S5).These effects were lost at subsequent measurements during follow up (week 30 and 54) (Fig. 2k, l). Due to the high number of tests (11 outcomes were examined) these results should be interpreted cautiously.

**Longitudinal assessment of neurofilament**
In the placebo arm an overall decrease in neurofilament levels was observed since the first follow-up by serial measurements in the serum (MD from baseline in serum pNfH at week 8: −174.95 ± 602.19; week 18: −289.35 ± 788.33; week 30: −482.29 ± 927.19; MD from baseline of serum NfL at week 8: −11.61 ± 96.07; week 18: −25.76 ± 93.46; week 30: −33.21 ± 101.57) (Tables S6, S7). The decrease of serum pNfH and NfL levels from baseline to week 18 that was found in the placebo group

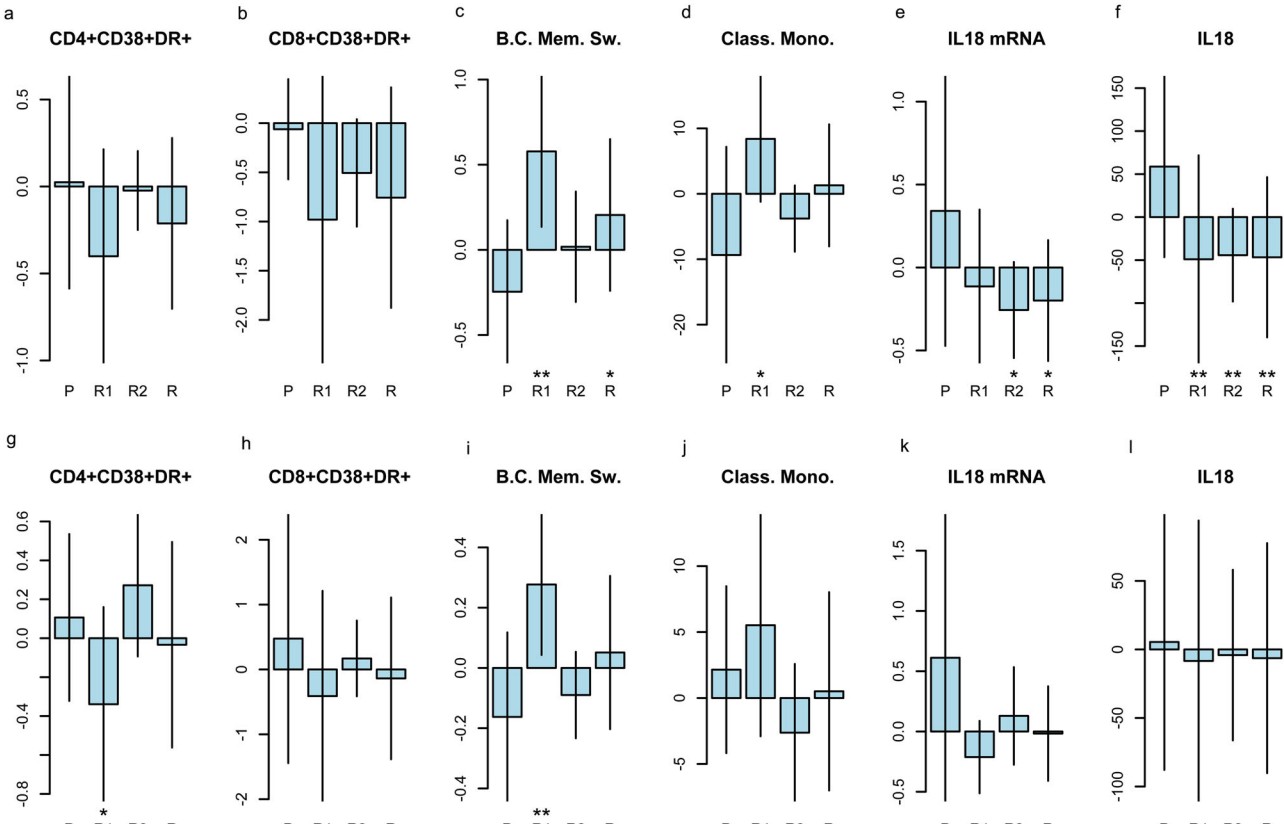

**Fig. 2 | changes from baseline in blood cells population and inflammasome across treatment arms (P = placebo, R1 = rapamycin 1 mg/m²/d, R2 = rapamycin 2 mg/m²/d, R = combined rapamycin arm).** The figure displays only a selection of the most interesting outcomes (55 cell subpopulation were examined and 11 inflammasome/cytokines, without accounting for multiple outcomes). In detail from left to right: changes from baseline to week 18 (**a**, *n* = 32 patients) and 30 (**g** *n* = 26 patients) in activated (CD38+, HLA-DR+) CD4 + T cells; changes from baseline to week 18 (**b**, *n* = 29 patients) and 30 (**h**, *n* = 21 patients) in activated (CD38 + , HLA-DR+) CD8 + T cells; changes from baseline to week 18 (**c** *n* = 22 patients) and 30 (**i**, *n* = 19 patients) in memory switched B cells (B.C. Mem. Sw.); changes from baseline to week 18 (**d**, *n* = 19 patients) and 30 (**j**, *n* = 19 patients) in classical monocytes (Class. Mono.); changes from baseline to week 18 (**e**, *n* = 27 patients) and 30 (**k**, *n* = 20 patients) in IL18 mRNA level (IL18 mRNA); changes from

baseline to week 18 (**f**, *n* = 49 patients) and 30 (**l**, *n* = 39 patients) of plasmatic IL18. Comparison were performed using linear regression models that include indicator variables for treatment arms as the independent variables. For the comparison of Rapamycin and placebo arms, a *P* value of 0.05 or less was considered to indicate statistical significance and uncertainty in results was expressed with the 95% confidence interval (CI). For the comparisons between Rapamycin 1 mg/m²/d or 2 mg/m²/d arms and the placebo arm, a *P* value of 0.025 or less was considered to indicate statistical significance and uncertainty in results was expressed with the 97.5% CI, to account for multiple arms comparison with the Bonferroni method. CIs were calculated based on the exact *t* distribution. All statistical tests were two-tailed. * means *p* < 0.05 for R and *p* < 0.025 for R1 and R2; ** means *p* < 0.01 for R and *p* < 0.005 for R1 and R2 arms. Error bars represent ± standard deviation. Source data are provided as a Source Data file.

was not observed in patients treated with rapamycin. This difference among treatment arms (pNfH MD 399.03, 95%CI 81.77 to 716.28; *p* = 0.015 and NfL MD 34.92, 95%CI 0.47 to 69.36; p = 0.047) was lost after treatment end (Tables S6, S7). Similar results were obtained from measurement of CSF pNfH and NfL at week 18 (pNfH MD 866.57, 95%CI −105.15 to 2345.72; *p* = 0.075 and NfL MD 4756.10, 95%CI 808.45 to 11531.5; *p* = 0.051, respectively) (Table S8).

### Other biological outcome measures
Monthly changes of selected biological outcome measures during and after treatment, across arms confirmed an increase of classical monocytes/CD14+ (MD 5.42, 97.5%CI 2.19 to 8.65, *p* = 0.0003) and memory switched B cells/CD45+ (MD 0.20, 97.5%CI 0.06 to 0.33, *p* = 0.0018) and a decrease of intermediate monocytes/CD14+ (MD −3.28, 97.5%CI −5.29 to −1.28; *p* = 0.0004) and IL-18 (MD −24.94, 97.5% CI −41.58 to −8.30; *p* = 0.009) in the rapamycin 1 mg/m²/day arm. The change from baseline to each time point of the phosphorylation of the S6RP was not different between rapamycin arms and placebo arms; this test was performed on a limited number of samples (Table S9). Changes from baseline to each time point in creatinine and albumin, CK, vitamin D, were not different in rapamycin and placebo arms.

### Secondary clinical outcomes
Absolute changes from baseline to each time point in ALSFRS-R total score in patients treated with rapamycin or placebo is showed in Table S10, whereas ALSFRS-R variation before, during and after treatment is showed in Table 4. Patients treated with rapamycin 1 mg/m²/d showed a mean monthly difference of 0.50 points in the ALSFRS-R total score with respect to placebo during treatment (97.5%CI −0.32 to 1.32; *p* = 0.172), and of −0.12 after treatment (97.5%CI −0.75 to 0.51; *p* = 0.671). The difference between the rapamycin 1 mg/m²/d group and the placebo group in the change in monthly variations from before to during treatment was 0.46 (97.5%CI −0.21 to 1.13; *p* = 0.174). Figures 3 and S3 show the mean and individual rate of decline of the ALSFRS-R total scores.

Correlation analyses was performed on changes in ALSFRS-r and neurofilament levels to investigate whether a relation existed between clinical and biological outcomes. In the placebo arm, an inverse correlation was found between the change (week 18−baseline) in serum and CSF neurofilament light levels and the change (baseline − week 18) in ALSFRS-R (Pearson's r coefficient: −0.41, 95% CI 0.09 to −0.74, *p* = 0.106 between serum NFL changes and ALSFRS-R, and r coefficient: −0.58, 95% CI −0.01 to −0.86, *p* = 0.049 between CSF NFL changes and

**Table 4 | Changes of ALSFRS-R monthly decline during and after treatment across treatment arms**

| Outcome | Period | | Arm | MD | CI | | p |
|---|---|---|---|---|---|---|---|
| ALSFRS-R Total score | Before treatment | Monthly variation | Placebo | −0.80 | −1.05 | −0.56 | 0.0000 |
| | During treatment | | Placebo | −1.39 | −1.89 | −0.89 | 0.0000 |
| | After treatment | | Placebo | −1.07 | −1.45 | −0.68 | 0.0000 |
| | Before treatment | | Rapamycin 1 mg | −0.77 | −1.00 | −0.53 | 0.0000 |
| | During treatment | | Rapamycin 1 mg | −0.89 | −1.40 | −0.37 | 0.0008 |
| | After treatment | | Rapamycin 1 mg | −1.19 | −1.58 | −0.79 | 0.0000 |
| | Before treatment | | Rapamycin 2 mg | −0.83 | −1.08 | −0.59 | 0.0000 |
| | During treatment | | Rapamycin 2 mg | −1.62 | −2.14 | −1.10 | 0.0000 |
| | After treatment | | Rapamycin 2 mg | −1.22 | −1.62 | −0.81 | 0.0000 |
| | Before treatment | Between-groups analysis: comparison with placebo (monthly variation) | Rapamycin 1 mg | 0.04 | −0.35 | 0.43 | 0.8240 |
| | | | Rapamycin 2 mg | −0.03 | −0.42 | 0.37 | 0.8676 |
| | | | Rapamycin | 0.01 | −0.29 | 0.30 | 0.9699 |
| | During treatment | | Rapamycin 1 mg | 0.50 | −0.32 | 1.32 | 0.1716 |
| | | | Rapamycin 2 mg | −0.23 | −1.06 | 0.60 | 0.5321 |
| | | | Rapamycin | 0.14 | −0.48 | 0.76 | 0.6604 |
| | After treatment | | Rapamycin 1 mg | −0.12 | −0.75 | 0.51 | 0.6706 |
| | | | Rapamycin 2 mg | −0.15 | −0.79 | 0.49 | 0.5948 |
| | | | Rapamycin | −0.14 | −0.62 | 0.34 | 0.5777 |
| | During treatment | Intra-group analysis: comparison with before treatment period (monthly variation) | Placebo | −0.58 | −1.05 | −0.12 | 0.0146 |
| | | | Rapamycin 1 mg | −0.12 | −0.60 | 0.36 | 0.6238 |
| | | | Rapamycin 2 mg | −0.78 | −1.27 | −0.30 | 0.0016 |
| | After treatment | | Placebo | −0.26 | −0.61 | 0.08 | 0.1327 |
| | | | Rapamycin 1 mg | −0.42 | −0.77 | −0.07 | 0.0177 |
| | | | Rapamycin 2 mg | −0.39 | −0.74 | −0.03 | 0.0336 |
| | During treatment | Between-groups analysis: comparison with placebo (difference in monthly variation compared to before treatment period) | Rapamycin 1 mg | 0.46 | −0.21 | 1.13 | 0.1742 |
| | | | Rapamycin 2 mg | −0.20 | −0.87 | 0.47 | 0.5581 |
| | | | Rapamycin | 0.13 | −0.45 | 0.71 | 0.6512 |
| | After treatment | | Rapamycin 1 mg | −0.16 | −0.65 | 0.33 | 0.5251 |
| | | | Rapamycin 2 mg | −0.12 | −0.62 | 0.37 | 0.6258 |
| | | | Rapamycin | −0.14 | −0.57 | 0.28 | 0.5125 |

Source data are provided as a Source Data file.

Average monthly variations before, during and after treatment for the placebo group, as well as the comparisons between arms, are shown. Comparisons were performed using segmented repeated measures linear mixed models. Three segments of time were analyzed: before treatment (from onset to baseline), during treatment (after baseline and up to week 18), and after treatment (after week 18). The dependent variables were the raw measurements of the outcomes, whereas the independent variables were: arm, time (months from baseline), period (before, during or after treatment), and all their pairwise and three-way interactions. A random intercept term was also used to account for repeated measurements over the same individual, as well as a random slope term was used to account for individual linear variations over time. Random intercept and random slope terms were kept in the model if they improved the overall goodness-of-fit of the model. For the comparisons between Rapamycin 1 mg/m²/d or 2 mg/m²/d arms and the placebo arm, a P value of 0.025 or less was considered to indicate statistical significance and uncertainty in results was expressed with the 97.5% confidence interval (CI), to account for multiple arms comparison with the Bonferroni method. For the comparison of Rapamycin and placebo arms, as well as for monthly variations and for the intra-group comparisons, a P value of 0.05 or less was considered to indicate statistical significance and uncertainty in results was expressed with the 95% CI. CIs were calculated based on the exact t distribution using the Satterthwaite's method for degrees of freedom. All statistical tests were two-tailed.

*MD* mean difference, *CI* confidence interval.

ALSFRS-R), indicating a decrease in neurofilament while increasing disease progression. Correlation analyses among the change (week 18−baseline) in serum and CSF neurofilament light levels and the change (baseline - week 18) in ALSFRS-R within rapamycin arm showed no correlation: Pearson's r coefficient was 0.18 (95% CI −0.50 to 0.17, $p = 0.308$) between ALSFRS-R and serum NFL change (w18−w0), and 0.32 (95% CI −0.65 to 0.10, $p = 0.131$) between ALSFRS-R and CSF NFL changes (w18−w0). Similar results were obtained with pNfH (Fig. 4). Comparison among correlation coefficients showed a nearly significant difference between rapamycin and placebo arm in serum NfL ($p = 0.0608$); the difference was less pronounced for pNfH ($p = 0.1429$).

There were no statistically significant differences between patients treated with rapamycin and placebo as far as PEG (19.0% in the placebo group, 14.3% in the rapamycin group, $p = 0.664$) or NIV

positioning (28.6% in the placebo group, 19.0% in the rapamycin group, $p = 0.468$) are concerned. During the study 7.1% of patients treated with rapamycin and 4.8% of patients in the placebo group had died ($p = 0.672$). There was only one IV in the placebo group. The most common cause of death was respiratory failure, accounting for three of the four deaths, a finding consistent with the natural history of ALS.

A post-hoc analysis on tracheostomy-free survival with last observation set on 31st December 2021, showed that 52.4% of patients treated with rapamycin and 61.9% of patients in the placebo group had died or underwent tracheostomy, not a statistically significant difference ($p = 0.356$) (Table S11, Fig. S4). There were no significant differences in the mean absolute change from baseline to each time point in the FVC (at week 18, MD 3.20, 97.5%CI −9.798 to 16.19, $p = 0.571$ for rapamycin 1 mg/m²/d group, and MD −4.57, 97.5% CI −17.56 to 8.42, $p = 0.419$ for rapamycin 2 mg/m²/d group)

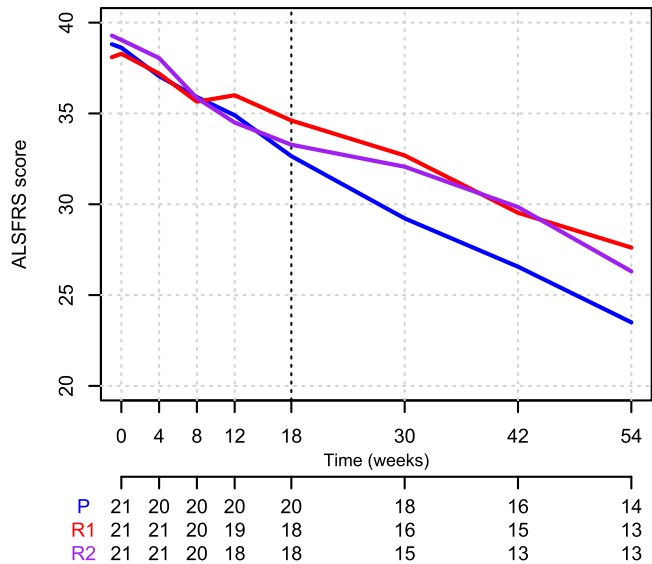

**Fig. 3 | Mean rates of decline in Amyotrophic Lateral Sclerosis Functional Rating Scale Revised (ALSFRS-R) total score (Intention to Treat population) of patients enrolled in RAP-ALS over the study (baseline to week 54) based on treatment arm allocation (red = R1, rapamycin 1 mg/m²/d, violet = R2, rapamycin 2 mg/m²/d, blue = P, placebo).** Source data are provided as a Source Data file.

(Table S12, Fig. S5). There were no significant differences in the mean absolute change from baseline to each time point in the ALSAQ40 scores (Tables S13, S14). ALSAQ40 main questions scores are represented in Fig. 5. Correlations among clinical outcome measures are presented in Table S15.

### Safety and drug adherence

A total of 23 over 42 individuals (55%) in the rapamycin group and 11 over 21 individuals (52%) in the placebo group had one or more AEs during the trial (Table S16). The total number of reported AEs was 23 for placebo arm, and 46 for the rapamycin arms. Severe AEs (SAEs) were 7 in the placebo group (30.4% of total AEs in that group), and 9 in the rapamycin groups (19.6% of total AEs in those groups) (Table S17). Among the totality of AEs, four caused treatment discontinuation (one in the placebo group and three in the rapamycin groups) (Table S17).

Individuals with SAEs were 19% both in the placebo and in the rapamycin groups (Table S18).

Events occurring at a greater frequency in the rapamycin group were primarily skin and subcutaneous tissue disorders (erythema, pruritus, rash, conjunctivitis, dermatitis, eczema), then gastro-intestinal disorders, injuries, respiratory disorders, headache and psychiatric disorders (Table 5).The majority of SAEs were represented by complications related to disease progression, such as dysphagia and hospital admissions to undergo PEG positioning, or respiratory failure/pneumonia. One subject committed suicide.

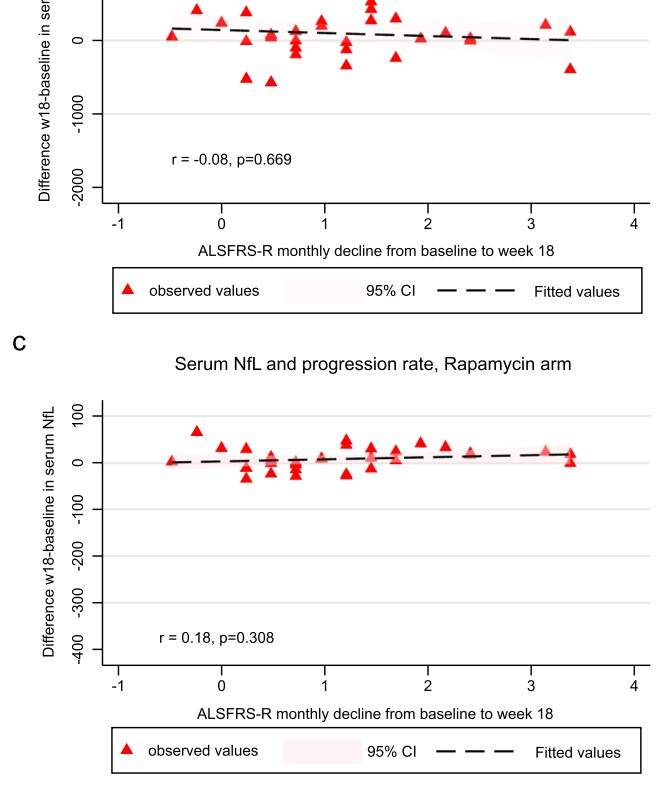

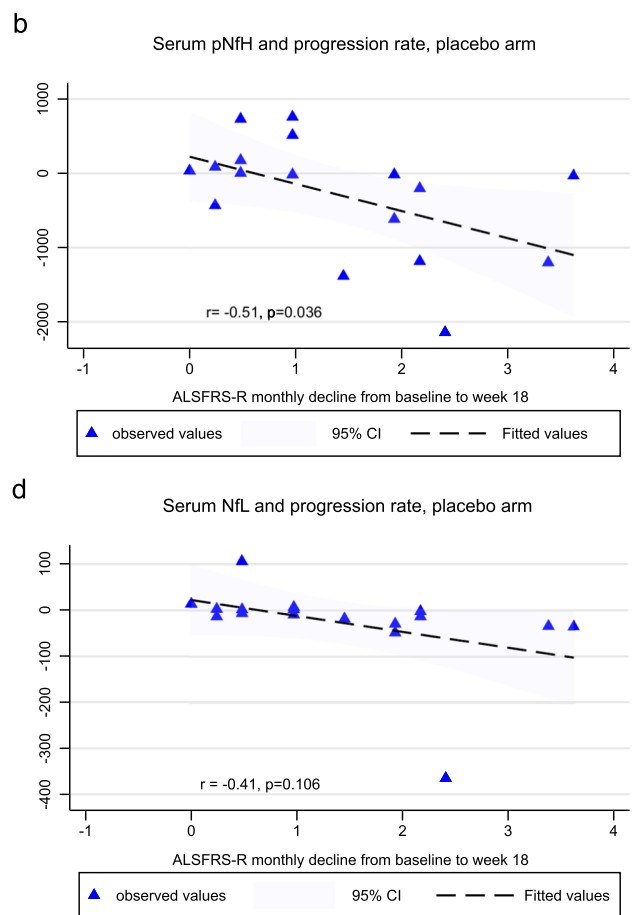

**Fig. 4 | Changes from week 18 to baseline in serum Neurofilament Light (NfL) and phosphorylated Neurofilament Heavy (pNfH) in relation to progression rate across treatment arms.** In detail from left to right, upper panels: changes from week 18 to baseline in serum pNfH in rapamycin (**a**) and placebo arm (**b**), in relation to progression rate calculated as the monthly decline in the ALSFRS-R from baseline to week 18. From left to right, lower panels: changes from week 18 to baseline in serum NfL in rapamycin (**c**) and placebo arm (**d**), in relation to

progression rate calculated as the monthly decline in the ALSFRS-R from baseline to week 18. Individual differences in neurofilament concentration between week 18 and baseline are plotted as colored symbols (Rapamycin arms in red; placebo arm in blue). The shaded areas represent the 95% confidence intervals around the model estimates. The lines and confidence intervals are drawn from the actual distributions of linear model fits. All statistical tests were two-tailed. Source data are provided as a Source Data file.

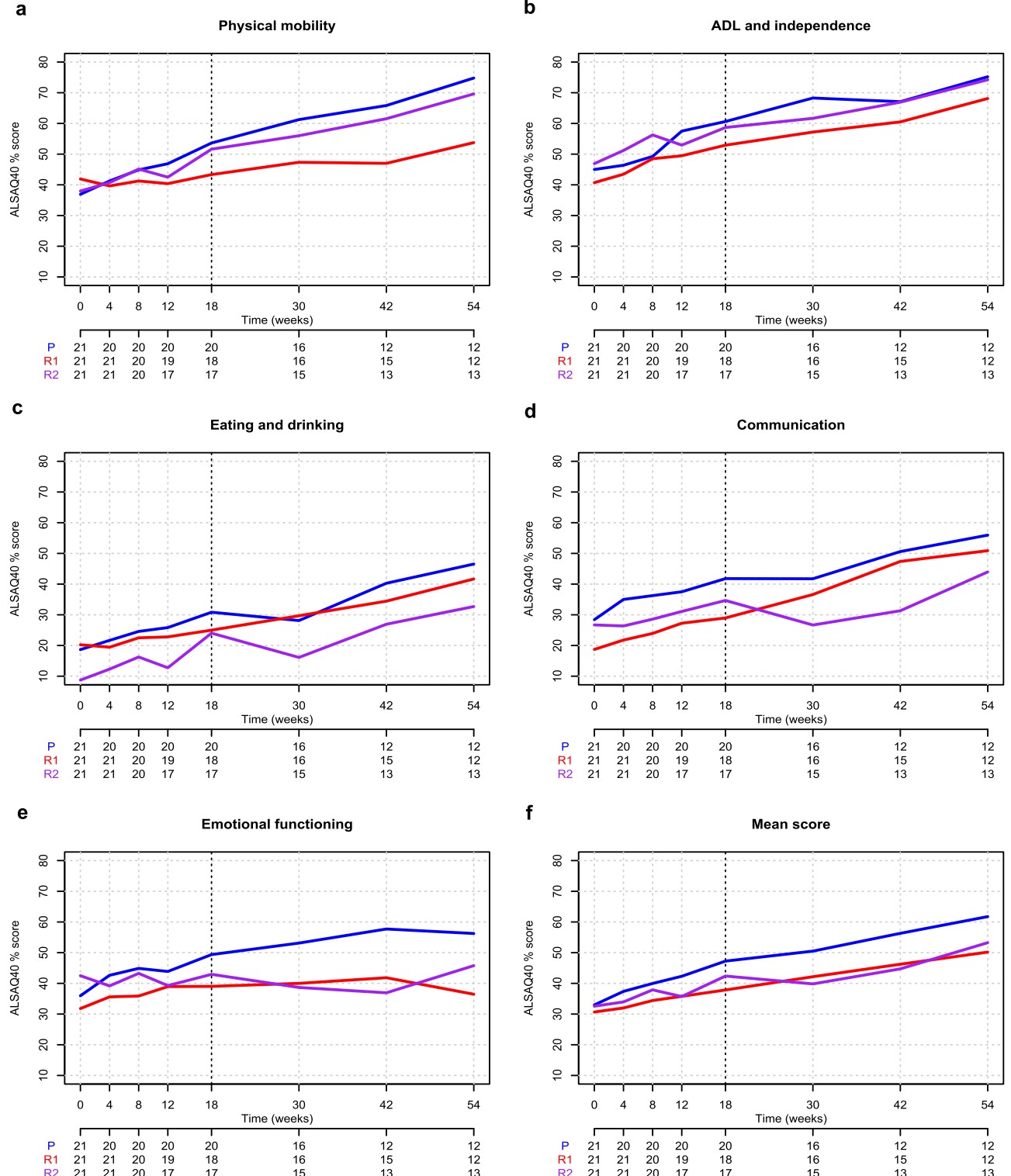

**Fig. 5 | Mean scores of ALS Assessment Questionnaire (ALSAQ40) from baseline to study end across treatment arms. A–E** These show the treatment-dependent mean scores of ALSAQ40 physical mobility, Activity Daily Living (ADL) and independence, eating and drinking, communication, and emotional functioning main questions, respectively, from baseline to study end. Panel F shows ALSAQ40 mean total score from baseline to study end (red = R1, rapamycin 1 mg/m²/d, violet = R2, rapamycin 2 mg/m²/d, blue = P, placebo; Intention to treat population). Source data are provided as a Source Data file.

Of note, there was a case of acute hepatitis probably related to the study drug, that occurred in a subject allocated to the rapa-mycin 2 mg/m²/day treatment arm, who had a very high peak of rapamycin at first blood dosage and for whom dose reduction applied. No permanent consequences had been reported from this event.

A total of 11% of the participants dropped out during the study treatment, 5% in the placebo group and 14.3% in the rapamycin group. During follow up time, 24% of patients in each group abandoned the study. Events leading to discontinuation of the treatment are pre-sented in Table S19. Data on adherence to the trial regimen are sum-marized in Table S20.

**Table 5 | Treatment-emergent adverse events**

| Treatment-emergent adverse events, MedDRA preferred term | PLACEBO (n = 21) | | Rapamycin 1 mg/m²/d (n = 21) | | Rapamycin 2 mg/m²/d (n = 21) | |
|---|---|---|---|---|---|---|
| | n | % | n | % | n | % |
| Blood and lymphatic system disorders | 0 | 0.0% | 1 | 4.8% | 0 | 0.0% |
| Depression and suicide/self-injury | 0 | 0.0% | 1 | 4.8% | 0 | 0.0% |
| Ear and labyrinth disorders | 1 | 4.8% | 0 | 0.0% | 0 | 0.0% |
| Eye disorders | 1 | 4.8% | 0 | 0.0% | 0 | 0.0% |
| Gastrointestinal disorders | 0 | 0.0% | 1 | 4.8% | 2 | 9.5% |
| General disorders and administration site conditions | 1 | 4.8% | 1 | 4.8% | 0 | 0.0% |
| Infections and infestations | 4 | 19.0% | 3 | 14.3% | 4 | 19.0% |
| Injury, poisoning, and procedural complications | 3 | 14.3% | 2 | 9.5% | 6 | 28.6% |
| Investigation | 0 | 0.0% | 1 | 4.8% | 0 | 0.0% |
| Metabolism and nutrition disorders | 1 | 4.8% | 0 | 0.0% | 2 | 9.6% |
| Musculoskeletal and connective tissue disorders | 2 | 9.6% | 0 | 0.0% | 0 | 0.0% |
| Nervous system disorders | 1 | 4.8% | 0 | 0.0% | 1 | 4.8% |
| Psychiatric disorders | 0 | 0.0% | 0 | 0.0% | 1 | 4.8% |
| Renal and urinary disorders | 2 | 9.6% | 0 | 0.0% | 0 | 0.0% |
| Renal and urinary disorders—Investigation | 1 | 4.8% | 0 | 0.0% | 0 | 0.0% |
| Respiratory, thoracic, and mediastinal disorders | 2 | 9.5% | 4 | 19.0% | 5 | 23.8% |
| Skin and subcutaneous tissue disorders | 1 | 4.8% | 5 | 23.8% | 6 | 28.6% |
| Vascular disorders | 1 | 4.8% | 0 | 0.0% | 0 | 0.0% |
| Not classified | 2 | 9.5% | 0 | 0.0% | 0 | 0.0% |

**Drug dosage assessment**

Plasma levels of rapamycin at different time points for each treatment arm are displayed in Fig. S6. While patients who were allocated to rapamycin 1 mg/m²/day treatment had a quite stable plasma dosage, patients allocated to the other treatment arm, during the first dosages presented an initial peak, with plasma levels above the upper limit of therapeutic range and also of the safety threshold (15 ng/ml), that next required drug re-dosing either to 1 mg/m²/day arm or to the minimal dose of 1 mg/day. Finally, examining CSF samples of patients at week 18 by LC-MS/MS a clear peak corresponding to Sirolimus was not detected neither in the CSF of treated patients, nor in the CSF of placebo (Fig. S7).

## Discussion

This clinical trial measured the biological, clinical and safety effects of rapamycin on patients affected by ALS. Unfortunately, the primary outcome measure could not be satisfied also due to the reduced number of samples that could be analyzed at week 18, to which the COVID-19 pandemic significantly contributed. Notwithstanding our expectations for an inevitable loss in samples at treatment end, this led to a final sample size of 50 on which the primary endpoint was assessed. Therefore, we could not demonstrate a significant effect of the study drug on Treg lymphocytes, and the question as to why some subjects did not show the expected Tregs increase in response to rapamycin remains.

Inter-individual variation amongst ALS patients in the ability to increase Tregs in response to treatment have been recently observed in other RCTs[18]. Since rapamycin effect on mTOR results from a delicate equilibrium between time and doses, many factors may affect the highly variable individual drug response including low bioavailability caused by liver and gut metabolism, genetic changes in the enzymes (mainly CYP3A4 and CYP3A5) involved in rapamycin metabolism, transportation and pharmacodynamics[19].

Furthermore, increased expression of the mTOR-interacting Raptor protein, increased phosphorylation of Akt, and activation of growth-related transcriptional factor AP-1 may induce rapamycin resistance[20], as well as mutant SOD1[21]. The complex and incompletely understood action of mTOR in both the regulation of lymphocyte differentiation and the enhancement of autophagy, may also explain why in a post-hoc analysis we did not find reduced Th1 markers in the treated group, besides the small sample analyzed; a previous study demonstrated that rapamycin yields an anti-apoptotic Th1/Tc1 effector phenotype by promoting autophagy[22]. In fact, mTOR blockade during human Th1/Tc1 cell generation induced autophagy, which promoted the survival of an anti-apoptotic T cell population that possesses the capacity to persist in vivo for prolonged intervals[22]. Finally, if rapamycin effect on immune response is dose-dependent, a high exposure to rapamycin may also cause inhibition of mTORC2 with detrimental immunologic and metabolic consequences[23]. This may explain the different results obtained on several immunological parameters in the group treated with rapamycin 1 mg/m²/d.

Among secondary biological outcome measures we observed changes on B and T cell subpopulations, monocytes, and on IL-18, which could be suggestive of a rapamycin-mediated effect on neuroinflammation in ALS, where misfolded protein aggregates may activate a cascade of events that drives chronic inflammation and secretion of proinflammatory cytokines, among which IL-18, that finally leads to tissue damage and cell death[24]. Nonetheless, these results should be interpreted cautiously and require confirmation in larger studies, considering that we did not apply corrections for multiple tests.

Together with the above mentioned limitation, the increase in classical monocytes and the reduction in intermediate monocytes in patients treated with rapamycin, is even more difficult to interpret because of the uncertainty on their role in ALS, and if monocyte subsets and activation profiles are altered depending on the stage of the disease (i.e., the changes are a response to disease and their changes suggest that the immune system becomes more activated as the disease progresses) or if they are instead pathogenetic deserves further studies[25–27]. The role of B cells in ALS has to be established yet, but increase in switched memory B cells, a population that differentiate into plasma cells upon reactivation[28], may suggest a compensatory mechanism of controlling inflammation by B cells. Overall, our study shows that there is a minor impact of the treatment on the measured outcomes given the large number of small group differences, but further studies should be performed to look at other biomarkers, such as Treg suppressor function and the best dose, to establish if rapamycin deserves to be pursued in larger ALS trials.

Serial measurements of neurofilament showed a constant decrease in placebo-arm patients, not observed in the treatment arms, where we found stable neurofilament levels. The different behavior of neurofilament in treatment and placebo arms may have several explanations since longitudinal behavior of neurofilament during ALS and/or in clinical trials is still being studied with conflicting results. Some studies found fast disease progression to be associated with neurofilament decrease over time[29], other observed an absence of correlation between a drug clinical effect and repeated neurofilament measurements[30,31], others correlated a rise of serum NfL over time to a fast-progressing disease course[32]. Differently from other trials we followed up patients with regular serum collection for 54 weeks, ensuring a relatively long period of observation and revealing an unusual trajectory in the placebo-arm patients, with a constant decrease for serum NfL and an initial steep decrease later diminished for serum pNfH. This may suggest that neurofilaments concentration is affected not only by the rate of neuronal degeneration but also by the phase of the disease

and the share of surviving MNs (i.e., in the presence of massive MNs degeneration, neurofilament release diminishes as a consequence of the decreased number of surviving MNs)[33]. As far as rapamycin arm is concerned, preclinical studies demonstrated that rapamycin reduces TDP-43 proteinopathy through autophagy enhancement and consequently relieves the pathogenic translational suppression of neurofilaments with increased protein synthesis[6,34]. This hypothesis is supported by the reversibility of dose-related biological effects upon rapamycin discontinuation. Therefore, neurofilament concentrations in relation with highly active neurodegenerative processes such as ALS may uncover different pathobiological mechanisms and should be interpreted cautiously in the setting of experimental drug testing[35]. In fact, we found an inverse correlation between disease progression and longitudinal change in NF only in the placebo group, whereas in rapamycin-treated patients there was no correlation between NF changes and disease progression.

This is a non-profit, exploratory study on rapamycin action in ALS patients, that was designed to investigate if this drug warrants further research in patients with ALS, considering also its safety profile. Rapamycin in combination with riluzole, was safe and well tolerated by ALS patients. AEs and SAEs were equally distributed between treatment and placebo arms, reassuring about safety, provided that a drug plasma dosage monitoring was performed. It is also reassuring that ~24% of the participants discontinued the trial, that is in line with recent data from other clinical studies[36]. Besides demonstrating that rapamycin is safe in patients with ALS, we found that rapamycin dosage of 1 mg/m²/day ensured a better stability of plasma dosages (never overcoming toxicity threshold) that only seldom required dosages adjustment due to safety concerns.

This study was not powered to test an effect on clinical measures, but we could not observe a slowing in the ALSFRS-R decline nor an effect on quality of life, even after excluding SOD1 patients from enrollment. If the effect on Treg cells and other immunological outcome measures was more evident for patients treated with rapamycin 1 mg/m²/day, the absence of data on Treg function requires further studies to ascertain what dose has the more beneficial immunological effect in ALS.

Given the reported detrimental effects in SOD1-ALS mice[11], however, next studies should be again cautiously restricted to non-*SOD1* ALS patients. A specific detrimental interaction between rapamycin and SOD1 may be hypothesized[21], reinforcing the concept that clinical and pathological heterogeneity of ALS may require personalized therapeutic strategies.

The main potentiality of this trial stands on immediate availability of the drug for patients use, should an effect be found by larger trials, and by the existence of newer and interesting molecules i.e., rapalogs, who offer the possibility to cross the blood brain barrier perhaps deserving further properties towards autophagy and neuroinflammation. In fact, we could not find the drug in the CSF of patients treated for 18 weeks, and although we cannot exclude rapamycin free fraction values below the detection limit of LC-MS/MS, it is probable that rapamycin cannot cross blood brain barrier. If this may be of secondary importance for the drug immunological effects, the concentration at which rapamycin or any rapalog are found able to penetrate the blood-brain barrier might be a matter of uttermost relevance for the sake of fostering autophagy[13].

Limitations of this early trial of rapamycin in ALS were the small number of participants and the short duration of treatment, that were mainly due to safety concerns. Furthermore, the expected effect size of the primary outcome measure was perhaps too optimistic and, together with the difficulties in obtaining blood samples at treatment end, this led to miss the primary outcome. The choice of the primary outcome measure, as a binary response, has to be acknowledged as the main drawback of this study, together with the fact that Treg function, rather than their number would have been relevant for treatment effect. Indeed, we did not plan to study Treg suppressive function instead of their number alone and we were not able to assess Treg suppressor function as a post-hoc analysis due to lack of available samples. Before further studies in ALS, assessment of Treg suppressor function in response to different doses of rapamycin would be critical. In addition, some data on the primary outcome were not measured due to the COVID-19 pandemic, which may introduce a risk of bias into the results. However, the double-blind design of the study and the random missing mechanisms should have limited this risk. Other drawbacks are represented by the imbalance of some factors at baseline (such as in edaravone treatment), and the exploratory nature of the clinical outcomes. Finally, the results on biological outcomes should be interpreted cautiously and require confirmation in larger studies, considering that we did not apply corrections for multiple tests.

In conclusion, this trial demonstrated that treatment with a low dose of rapamycin is safe in patients with ALS, but it failed to demonstrate an effect of the drug on Treg cells. Further trials focused on different outcome measures are necessary to better understand rapamycin biological and clinical effects in ALS.

## Methods

### Study design

A randomized, double-blind, placebo-controlled trial was conducted at seven Italian ALS referral centers from 2017 through 2020. The trial was conducted in accordance with the Good Clinical Practice guidelines of the International Council for Harmonization of Technical Requirements for Registration of Pharmaceuticals for Human Use (ICH) and the ethical principles of the Declaration of Helsinki, as amended by the 64th WMA General Assembly, Fortaleza, Brazil, in October 2013. The study complies with ICMJE guidelines on reporting.

Protocol (EUDRACT 2016-002399-28 registration: 31st May 2016) approval was provided for the coordinating center and all trial sites by Ethical Committee of the coordinating center (Comitato Etico Provinciale di Modena) on 23th May 2017 (file number 95/17) and by AIFA (Agenzia Italiana del Farmaco) on 14th July 2017. All the participants provided written informed consent before screening (first and last patient enrollment: 05/10/2017 to 02/01/2020). The trial was a non-profit trial, financed by ARISLA (Fondazione Italiana di Ricerca per la SLA) with the "2015 AriSLA Ice Bucket Call for Clinical Projects". Pfizer Inc. provided the active drug. Participants received no compensation.

The study promoter was Azienda Ospedaliero-Universitaria (AOU) di Modena. The trial design, data analysis, and manuscript development were shared by the Steering Committee of the Study represented by all the local PIs (Supplementary Appendix, Section 2). The authors were responsible for writing the manuscript and making the decision to submit it for publication. Confidentiality agreements were set up between the authors and ARISLA. Data collected at each trial site were input in an online case report form (CRF) written in PHP version 7.2, jquery version 3.4.1 and PostgreSQL version 9.5 as a database backend, set up and managed by the Coordinator Center at AOU, Modena and only accessible to RAP-ALS investigators through password protected access.

An independent data and safety monitoring board was established at trial beginning and periodically reviewed unblinded safety data during the trial (Acknowledgements section). Statistical analyses were performed by the Unit of Statistical and Methodological Support to Clinical Research, AOU, Modena, Italy. All the authors guarantee for data completeness and accuracy, and for adherence to RAP-ALS study protocol (available at http://www.nature.com/ as supplementary material of the present article and already published[17]).

### Trial participants

The trial enrolled patients diagnosed with definite, clinically probable or probable with laboratory support ALS according to revised El

Escorial criteria who presented ALS symptoms onset not earlier than 18 months before screening. Inclusion criteria encompassed age between 18 and 75 years old, a forced vital capacity (FVC) exceeding 70% of the predicted value for sex, age, height, and weight, a Body Mass Index above 18 and a body weight over 50 kg, use of riluzole at a stable dose for at least 30 days before screening. Exclusion criteria covered a wide range of diseases and conditions that would make rapamycin use or immunosuppression contraindicated. Patients with known SOD1 mutation or with FALS and family members carrying SOD1 mutation were to be excluded as well, based on contrasting evidence of Rapamycin action in SOD1 models of ALS,[37,38] quite the reverse on previous studies on models linked to TDP43 pathology[14,39]. Inclusion and exclusion criteria are fully described in the published protocol[17].

### Randomization and masking

Eligible participants were randomly assigned in three treatment arms with a 1:1:1 ratio to receive rapamycin 1 mg per square meter ($m^2$) of body surface area a day (1 mg/$m^2$/day) (21 patients), rapamycin 2 mg/$m^2$/day (21 patients), or placebo (21 patients). The randomization schedule was computer generated by an unblinded statistician using STATA software (StataCorp. 2017. Stata Statistical Software: Release 15). Randomization was stratified by rate of disease progression as measured by monthly decline of the Revised ALS Functional Rating Scale (ALSFRS-R) from onset to screening visit, with a cut off set at </≥0.7. Euromed Clinical Supply Services (ECLISSE) (Cantù, Como, Italy; http://www.css.euromed.it/en/) prepared the active formulation and the placebo complying the Good Manufacturing Practices of the European Union for active pharmaceutical ingredients and ICH Q7A guidelines. Trial drug was dispensed in kits with random four-digit identification numbers. Kits were sent in sequence to sites as each new participant was enrolled. Treatment under investigation and placebo were made indistinguishable to patients and neurologists.

### Procedures

Treatment was administered orally, in the morning, at fast. Patients received 4 bottles, each containing 15 tablets of active drug or placebo depending on the assigned treatment arm, every 2 weeks, for a planned duration of 18 weeks. Rapamycin plasma levels were regularly measured by high-performance liquid chromatography with tandem mass spectrometry (LC–MS/MS) and made known only to an independent monitor, who input rapamycin levels on a separate eCRF, allowing dose reduction if rapamycin plasma levels exceeded 12 ng/mL. A dose reduction could be asked also by caring neurologist directly through eCRF in case of adverse events (AEs) or reactions that, on clinical judgment, could be attributed to the study drug. After treatment end patients had to be followed up for further 36 weeks (https://www.clinicaltrialsregister.eu/ctr-search/trial/2016-002399-28/IT; ClinicalTrials.gov, NCT03359538).

### Outcomes

The primary efficacy outcome was the proportion of positive response (Tregs number increase of at least 30%) at treatment end (18 weeks) with respect to baseline, in patients treated with rapamycin compared to the placebo arm. This difference was established based on a previous study demonstrating that slowly progressing ALS patients presented a number of Tregs that was equal to healthy controls, whereas fast progressors had 31% fewer Tregs[40]. Since Tregs % were demonstrated to be inversely correlated with the rate of disease progression,[41] we considered a "positive response" as an increase of the proportion of Tregs by at least 30% at treatment end.

Secondary outcomes were:

I. Assessment of rapamycin safety and tolerability through documentation of the occurrence of any AEs, changes on clinical examination including vital signs and weight, and laboratory examinations (biochemistry, hematology and urinalysis) that were registered throughout the study duration. Symptoms consistent with disease progression, were recorded as AEs.

II. Biological outcomes, assessed as the change from baseline to week 8, 18, 30, 54, comparing rapamycin and placebo arms, of the following biological variables: a) activation and homing capabilities of different T, B, NK cell subpopulations; b) relative expression of inflammasome genes and its activation status; c) phosphorylation of the S6 ribosomal protein (S6RP); d) plasma/CSF neurofilament heavy/light chain protein; e) creatinine and albumin, CK, vitamin D; f) the assessment of rapamycin in CSF samples at week 18 by LC-MS/MS.
   Laboratory methods are explained in Supplementary Appendix, Section 3.1.

III. Clinical outcomes through comparison between placebo and treatment arms of: a) the changes in ALSFRS-R from baseline to weeks 4, 8, 12, 18, 30, 42, 54; b) overall survival from randomization to date of documented death or tracheostomy; c) survival rate at weeks 18, 30, 42, and 54; d) respiratory muscle function as assessed by FVC score from baseline to weeks 4, 8, 12, 18, 30, 42, 54 (Supplementary Appendix, Section 3.2).

IV. Quality of life, measured through absolute and relative change from baseline in ALSAQ-40 at week 4, 8, 12, 18, 30, 42 and 54 comparing treatment and placebo arms.

### Statistical analysis

Sample size was calculated using data from an Italian study showing that ALS patients have a decreased number of circulating Treg% (mean ± SD: 2.1 ± 0.7) if compared to healthy controls (2.6 ± 0.6), except for slow progressors[40]. Considering normal values of total Treg of 71.5 ± 17/mmc on a normal total lymphocytes count between 1000 and 4500/mmc, slowly progressing ALS patients presented a number of Tregs that was equal to healthy controls, whereas fast progressors had 31% fewer Tregs[40]. Since Treg % were demonstrated to be inversely correlated with the rate of disease progression,[41] we considered a "positive response" as an increase of the proportion of Tregs by at least 30% at treatment end. The null hypothesis was that Rapamycin could not significantly increase the proportion of positive responses in treated patients at week 18, compared to baseline and to placebo group. The alternative hypothesis was that Rapamycin could determine a positive response in at least 50% of treated patients compared to a maximum 5% of patients in the placebo group. The study was designed to reject the null hypothesis with an alpha error of 0.025 (in order to take into account multiple arms comparisons) and a power of 0.80. With a 1:1:1 randomization ratio among the three arms we calculated that 54 participants would provide 80% power to detect a 30% difference in the percentual of circulating Treg in at least 50% of treated patients versus less than 5% in the placebo group, using a chi-square test without any correction at a two-sided alpha level of 0.025. The study was planned to reach a sample size of 63 patients taking into consideration a possible 15% of drop out.

Safety analyses were performed including all patients who received at least one tablet of rapamycin or placebo. All AEs, SAEs, and AEs leading to treatment discontinuation were recorded according to ICH Guidelines, listed, and compared in the treatment arms at any follow-up visit and at the end of the study.

The primary population for analyses was the ITT population, which included all the participants who received at least one tablet of the investigation drug. PP analysis was performed after excluding patients as per major protocol deviations (i.e., patients who took <80% therapy) from the above-mentioned population. Descriptive statistics comparing the two groups of rapamycin treatment and placebo was performed using mean and standard deviation for continuous variables, counts and percentages for categorical variables. Immune response to rapamycin was analyzed as the difference in positive

response to rapamycin (mean Tregs increase exceeding 30%) between the placebo group and the rapamycin groups Results were expressed as the relative risk (RR) comparing treatment arms. The comparison was carried out with a chi-square test without any correction. Mean absolute differences from baseline to week 18 and other time points among treatment arms for S6RP phosphorylation, of different T, B, NK cell subpopulations, of biomarkers, inflammasome, cytokines, were calculated and compared using linear regression models that include indicator variables for treatment arms as the independent variables. Results were expressed as the mean difference (MD) comparing treatment arms. Correlations between numerical variables were calculated with the Pearson's linear correlation coefficient. The time to death, tracheostomy or permanent ventilation, were compared by using the log-rank test. Adjustment for main prognostic or unbalanced factors (namely sex, months from symptoms onset, ALSFRS-R slope at baseline and edaravone treatment) has also been performed using logistic or linear regression models. Results of logistic models were expressed as the OR comparing treatment arms.

A segmented repeated measures linear mixed model analysis was carried out to assess whether the average monthly variation from baseline in selected numerical outcomes was different amongst treatment arms in two segments of time: during the treatment (after baseline and up to week 18), or after the treatment (after week 18). The dependent variables were the raw measurements of the outcomes, whereas the independent variables were: arm, time (months from baseline) × period (during or after treatment) interaction, and arm × time × period interaction. A random intercept term was also used to account for repeated measurements over the same individual, as well as a random slope term was used to account for individual linear variations over time. Random intercept and random slope terms were kept if they improved the overall goodness-of-fit of the models (assessed using the Akaike information criterion). Results of this analysis were expressed as: the monthly variation of outcomes in the placebo group; the MD in the monthly variations, comparing treatments arms vs placebo. Both these quantities were reported for the two segments of time (during and after treatment). In the analysis of average monthly variation of ALSFRS-R, three segments of time were analyzed: before, during and after treatment, by considering two additional timepoints such as the prebaseline screening visit and the disease onset. At disease onset, the ALSFRS-R was assumed to be equal to 48 for all patients. The repeated measures linear mixed model for ALSFRS-R included arm, time (months from baseline), period (before, during or after treatment), and all their pairwise and three-way interactions. The intra-group differences comparing ALSFRS-R monthly variations in the periods during and after treatment with those occurred before treatment were also reported for each arm, as well as the comparison between treatment arms and placebo in these quantities. A post hoc analysis on tracheostomy-free survival from baseline to 31st December 2021 was performed.

For the comparisons between Rapamycin 1 mg/m²/d or 2 mg/m²/d arms and the placebo arm, a $P$ value of 0.025 or less was considered to indicate statistical significance and uncertainty in results was expressed with the 97.5% confidence interval (CI), to account for multiple arms comparison with the Bonferroni method. For the comparison of Rapamycin and placebo arms and, when applicable, for all other statistical measures, a $P$ value of 0.05 or less was considered to indicate statistical significance and uncertainty in results was expressed with the 95% CI. CIs for linear models were calculated based on the exact t distribution. For linear mixed models, the Satterthwaite's method for degrees of freedom was used. All statistical tests were two-tailed. Missing values were excluded from the analyses and the number of analyzed individuals in each arm was reported. Missing values were due to Covid-19 pandemic, and, for secondary biological outcome measures, also to the reduced amount of material of biological samples for difficulty in blood withdrawal; for FVC they were due to

increasing patients' difficulties to perform spirometry with advancing respiratory or bulbar impairment.

Analyses were performed using STATA software, version 15 (StataCorp. 2017. Stata Statistical Software: Release 15. College Station, TX: StataCorp LLC) and R software, version 3.6.3 (The R Foundation for Statistical Computing, Wien).

### Reporting summary
Further information on research design is available in the Nature Portfolio Reporting Summary linked to this article.

## Data availability
The data that support the findings of this study are available from the corresponding author (jessica.mandrioli@unimore.it) to external researchers who provide methodologically sound scientific proposals and whose proposed use of the data has been approved by an independent review committee identified for this purpose. All requests will be reviewed by corresponding author and Steering Committee of the Study and response to requests will be given in 2 months. A materials transfer and/or data access agreement with the study promoter will be required for accessing shared data. However, individual participant data will not be available because informed consent did not explicitly include this. Source data are provided as Source Data files with this paper. The study protocol, including the statistical analysis plan has been uploaded in the Supplementary Information file. Source data are provided with this paper.

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

## Acknowledgements

We thank all the RAP-ALS investigators group (see supplementary material). We thank the participants in the RAP-ALS trial and their families and caregivers, without whom this trial would have not been possible; Dr. Graziella Filippini, Dr. Ettore Beghi, Dr. Lawrence Korngut and Dr. Paola Minghetti, members of the data and safety monitoring board; Neurobiobank of Modena; Telethon Genetic BioBank (GTB12001D) and the Eurobiobank network; European Reference Network for Neuromus-cular Diseases (ERN EURO-NMD); CRO High Research- Evidenze Group.

## Author contributions

J.M., A.Co., R.D.A., M.P. had the idea for the study, did the literature searches and designed the study. E.Z., I.M., N.F., G.G., C.L., F.G., C.T., L.M., F.D.M., A.S., G.S., A.F., G.L., E.D.B., C.C., G.M., A.Ch., and A.Ca. were co-investigators, were responsible for patients recruitment, treatment, biological samples collection, data input in eCRF. Clinical oversight of the study was undertaken by J.M. and supported by R.D.A. The RAP-ALS Investigators provided feedback on the protocol and oversaw site recruitment and data collection for their respective sites. J.M., R.D.A., A.Co., E.Z., I.M., R.V., F.B. were responsible for development of protocols, site training, and validation and creation of the eCRF and database. The study conduct was overseen by J.M. and R.V., supported by R.D.A. and a local management team comprising I.M., E.Z., N.F. A.Co., S.D.B., M.P. developed standard operating procedures for immunological analyses. S.D.B., C.S., D.L.T., M.P., under the supervision of A.Co., performed immunological analysis of biological samples. Acquisition of the financial support for the project leading to this publication was performed by J.M., A.Co. and R.D.A. with participation of A.C., C.C., C.L., G.S., E.D.B., L.M. Project administration was performed by J.M., R.D.A. and A.Co. The final analysis of all the primary and secondary outcomes was undertaken by F.B. under the supervision of R.D.A., and both veri fied the raw data. The manuscript was drafted by J.M., E.Z., F.B., S.D.B. All authors had access to all the study data and read, contributed to, reviewed and approved the submission of the manuscript for final publication.

## Funding

The study was supported by ARISLA (Fondazione Italiana di Ricerca per la SLA) (FGCR02/2015 to J.M.) and Pfizer Inc. (Pfizer provided the drug free of charge) (grant no. 53232941, program title: "Wi211892 Rapamycin treatment for Amyotrophic Lateral Sclerosis" to J.M.). The funders of the study were not involved in protocol design, data collection, statistical analysis, data interpretation, writing of the report and the decision to submit this article.

## Competing interests

J.M. reports receiving advisory board fees from Biogen, Amylix and Italfarmaco, grant support from Roche, and grant support from Pfizer

(active study drug for this study by grant number Wi211892 to J.M.). ACh received consulting fees from Biogen, Cytokinetics, Amylyx. ACa reports receiving advisory board fees from Biogen and Amylix, and grant support from Cytokinetics. GL reports scientific advisory for CSL Behring, Biogen Inc, Vertex Pharmaceuticals Incorporated, Chromocell Corporation, Janssen Pharmaceuticals, Inc, Lilly, and the Bracco Group. C.L. has served as a scientific consultant for Mitsubishi Tanabe Pharma Europe, Cytokinetics, Neuraltus, and Italfarmaco. R.D.A., E.Z., S.D.B., F.B., I.M., C.S., D.L.T., R.V., N.F., G.G., M.P., F.G., C.T., L.M., F.D.M., A.S., G.S., A.F., E.D.B., C.C., G.M., A.Co. declare no competing interests. Disclosure forms provided by the authors are available with the full text of this article.

## Additional information

[1]Department of Biomedical, Metabolic and Neural Sciences, University of Modena and Reggio Emilia, Modena, Italy. [2]Department of Neurosciences, St. Agostino-Estense Hospital, Azienda Ospedaliero-Universitaria di Modena, Modena, Italy. [3]Unit of Statistical and Methodological Support to Clinical Research, Azienda Ospedaliero-Universitaria, Modena, Italy. [4]Department of Medical and Surgical Sciences for Children and Adults, University of Modena and Reggio Emilia, Modena, Italy. [5]Neurosciences PhD Program, University of Modena and Reggio Emilia, Modena, Italy. [6]Clinical and Experimental Medicine PhD Program, University of Modena and Reggio Emilia, Modena, Italy. [7]Department of Life Sciences, University of Modena and Reggio Emilia, 41125 Modena, Italy. [8]NEuroMuscular Omnicenter, Serena Onlus Foundation, Milan, Italy. [9]Istituto Maugeri IRCCS Milano, Milan, Italy. [10]ALS Centre, Neurologic Clinic, Maggiore della Carità University Hospital, Novara, Italy. [11]Department of Neurosciences, University of Padua, Padua, Italy. [12]Centro Regionale Specializzato Malattie del Motoneurone, Azienda Ospedale Università di Padova, Padua, Italy. [13]3rd Neurology Unit and ALS Centre, IRCCS 'Carlo Besta' Neurological Institute, Milan, Italy. [14]Department of Neurosciences, Rehabilitation, Ophthalmology, Genetics, Mother and Child Disease, Ospedale Policlinico San Martino, Genova, Italy. [15]'Rita Levi Montalcini' Department of Neurosciences, ALS Centre, University of Turin and Azienda Ospedaliero Universitaria Città della Salute e della Scienza, Turin, Italy. [16]National Institute for Cardiovascular Research, via Irnerio 48, 40126 Bologna, Italy. [17]These authors contributed equally: Jessica Mandrioli, Elisabetta Zucchi, Sara De Biasi, Federico Banchelli. [18]These authors jointly supervised this work: Roberto D'Amico, Andrea Cossarizza. ✉e-mail: jessica.mandrioli@unimore.it

