## [Peer Review File · Nature Communications]

Randomized, double-blind, placebo-controlled trial of rapamycin in amyotrophic lateral sclerosis (RAP-ALS study)Editorial Note: This manuscript has been previously reviewed at another journal that is not operating a transparent peer review scheme. This document only contains reviewer comments and rebuttal letters for versions considered at *Nature Communications*.

Reviewers' Comments:

Reviewer #1:

Remarks to the Author:

The paper presents the results of a randomized clinical trial of two doses of rapamycin in ALS patients. The paper presents a very large number of analyses comparing the treatment groups.

1) General: While I appreciate that you accounted for multiple comparisons, I believe that reporting the actual p-value and stating that the corrected type I error rate would be 0.025 is a better approach. The problem is the expected relationship between the confidence interval and p-value is not maintained if there is no consistency between the approaches. As an example, on lines 251 and 252, the 95% confidence intervals exclude 0 and the p-values are larger than 0.05. This would also reduce the number of p-values equal to 1.00 reported in the tables.

2) General: The paper includes caveats associated with multiple testing, but I believe that the main conclusion of the paper seems to be that there is a minor impact of the treatment on the measured outcomes given the large number of small group differences.

3) General: Across all of the analyses, I think there needs to be more description of the amount of missing data, the reasons for missingness, and any potential bias introduced due to missing data. It seems that missing values were removed using complete case analyses in all analyses, but there is no discussion of the missing data mechanism or any potential bias associated with this approach for handling missing data in this manner.

4) Line 226: I think the second RR refers to the 2mg/m²/d group.

5) Line 265: The reported difference is for the comparison of the placebo to the combined rapamycin group. Why is the combined group used here?

6) Line 276/Table S6-S7: Why is the standard deviation for the change in NfH and NfL much larger in the placebo group in weeks 8, 18, and 30? Is there one subject with a very large change?

7) Figure 1: How were the subjects who died incorporated in any of the analyses?

8) Table 4: I believe that the treatment effect in this model would be the difference in the change in the mean difference during treatment and before treatment comparing the rapamycin groups to the placebo group. I do not believe that is shown in this table. Could you more clearly show this difference?

9) Line 336: It is not clear why this is considered a trend toward a slower decline given the width of the confidence interval.

Reviewer #2:

Remarks to the Author:

The authors responded to previous reviews. The phase 2 trial did not reach its primary outcome measure of increasing t reg number.

The authors were not able to assess t reg suppressor function.

it is not clear that the conclusions of finding best dose hold. Before further studies in ALS, knowing whether t reg suppressor function is improved would be critical.

While an explanation is provided for choosing 30% increase in t reg number, this isn't clear why the number would be relevant- rather than function.

it may be that rapamycin should not be pursued further in als - or that there is need for another phase 2a trial to look at dose and other biomarkers (t reg suppression, etc).

Modena, 6th June 2023

Dear Professor,

On behalf of RAP-ALS investigators, I'm asking to consider the revised version of our article entitled "**Randomized, double-blind, placebo-controlled trial of rapamycin in amyotrophic lateral sclerosis (RAP-ALS study)**" for publication in Nature Communications as an original article.

We are very grateful for the time and the comments received by reviewers that allowed us to improve the manuscript.

Here, we respond to all the reviewers' remarks and we attach a clean and track-changes version of the manuscript.

We would be honored to have this opportunity to submit the paper to an eminent journal like yours.

We think that our manuscript is suitable for "Nature Communications", because of its findings and implications in such an emergent neurodegenerative disease as ALS especially for further trials.

We believe that the implications of our results can contribute to understand the effect of rapamycin administration in person with ALS. No randomized clinical trial with rapamycin was ever conducted in humans with ALS.

You can find below our point-to point answers to reviewers' comments.

REVIEWERS' COMMENTS TO THE AUTHORS

Reviewer #1:

Reviewer #1 (Remarks to the Author):

The paper presents the results of a randomized clinical trial of two doses of rapamycin in ALS patients. The paper presents a very large number of analyses comparing the treatment groups.

1) *General: While I appreciate that you accounted for multiple comparisons, I believe that reporting the actual p-value and stating that the corrected type I error rate would be 0.025 is a better approach. The problem is the expected relationship between the confidence interval and p-value is not maintained if there is no consistency between the approaches. As an example, on lines 251 and 252, the 95% confidence intervals exclude 0 and the p-values are larger than 0.05. This would also reduce the number of p-values equal to 1.00 reported in the tables.*

We agree with the reviewer and we modified the reporting of those results. In the revised manuscript, we reported the actual (not corrected) p-values, stating throughout the main text and the supplementary material, where applicable, that: for the comparisons between Rapamycin 1 mg/m²/d or 2 mg/m²/d arms and the placebo arm, a P value of 0.025 or less was considered to indicate statistical significance and uncertainty in results was expressed with the 97.5% confidence interval, to account for multiple arms comparison with the Bonferroni method; for the comparisons between Rapamycin arm and the placebo arm, a P value of 0.05 or less was considered to indicate statistical significance and uncertainty in results was expressed with the 95% confidence interval. We also thank the reviewer for highlighting the need for consistency between p-values and confidence intervals. In the previously submitted analysis of average differences from baseline, the confidence intervals

for the mean differences were calculated with a method which was slightly different from the one used for p-values, due to a typo in the R software codes used for generating the results. In the revised manuscript, we have made the two approaches consistent, and the methods for calculating confidence intervals and p-values were described in the statistical methods section (exact t distribution for the analysis of changes from baseline at each week; exact t distribution with Satterthwaite's method for degrees of freedom for repeated measures analysis).

2) *General: The paper includes caveats associated with multiple testing, but I believe that the main conclusion of the paper seems to be that there is a minor impact of the treatment on the measured outcomes given the large number of small group differences.*

We agree with the reviewer and underlined this consideration ("Our study shows that there is a minor impact of the treatment on the measured outcomes given the large number of small group differences") on page 9, lines 438-442 of the main file, TC version.

o3) *General: Across all of the analyses, I think there needs to be more description of the amount of missing data, the reasons for missingness, and any potential bias introduced due to missing data. It seems that missing values were removed using complete case analyses in all analyses, but there is no discussion of the missing data mechanism or any potential bias associated with this approach for handling missing data in this manner.*

Missing data were not filled in, and they were treated as such. All the data analyses presented in the study consider only individuals with non-missing data, and missing observations are excluded from the reported results. For each analysis on primary and secondary outcomes, the number of analysed individuals in each arm is reported. We acknowledge that this method can introduce a risk of bias in results, but the randomized, double blind study is the better design, to control for this possible bias.

For the primary outcome, missing data were due to lack of samples due to Covid-19 pandemic. As far as secondary biological outcome measures, together with this last cause for missingness, some measurement were not performed also due to the reduced amount of material of biological samples (due to difficulty in blood withdrawal). Among clinical secondary outcomes, ALSFRS-R was missing at week 54 for only one subjects, and ALSAQ40 for only four subjects. FVC missing values were more frequent due to the fact that with increasing bulbar or respiratory impairment, patients can't be able to perform spirometry (for instance due to the lack of capacity to hold the mouthpiece), and this is a common issue in ALS clinical trials.

Based on our judgment, there were only random differences in the missing data mechanism between treatment and placebo arms, given that the study was double-blinded and that the reasons for missingness are not related to treatment allocation or to outcome measurements.

We added how we dealt with missing values in the methods section (page 15, lines 728-732 of the main file, TC version); we added a note on missing values in the tables of the supplementary files, and we added a comment in the discussion (page 11, lines 518-521 of the main file, TC version).

4) *Line 226: I think the second RR refers to the 2mg/m2/d group.*

We corrected the typo on page 5, line 226 of the main file, TC version.

5) *Line 265: The reported difference is for the comparison of the placebo to the combined rapamycin group. Why is the combined group used here?*

We changed this part by reporting the comparison between placebo and each treatment group (page 6, lines 266-271 of the main file, TC version).

6) *Line 276/Table S6-S7: Why is the standard deviation for the change in NfH and NfL much larger in the placebo group in weeks 8, 18, and 30? Is there one subject with a very large change?*

There was a single subject in the placebo group with a very large change in serum NfH and NfL, which inflates the standard deviation in this arm. To make an example, in serum NfL at week 18, this subject has a -365 pg/ml variation from baseline, compared to an average reduction of the other patients in the placebo group equal to -4.65 pg/ml. The standard deviation in the placebo group, after excluding this subject, would be equal to 34.1 pg/ml, which would be very similar to the standard deviation observed in the rapamycin arms. By excluding this subject from analyses, the MD between rapamycin and placebo arm at week 18 with respect to baseline would be 13.71 (p=0.109) for serum NfL and 283.11 (p=0.048) for pNfH, not substantially different from values obtained including that subject: 34.92 (p=0.047) for NfL and 399.03 (p=0.015) for serum pNfH. We added a note in table S6-8 on this.

7) *Figure 1: How were the subjects who died incorporated in any of the analyses?*

Subjects who died were excluded from the analysis of the primary and secondary biological and clinical outcomes, if their outcome values were not measured at the time-points of interest. The between-groups comparison of time-to-death was carried out with the log-rank test and reported in the Supplementary Materials, Table S11. For other time-to-event outcomes, individuals who died were treated as being censored observations.

All the data used for Figure 1 can be found in supplementary material (Table S19) and in the source data.

8) *Table 4: I believe that the treatment effect in this model would be the difference in the change in the mean difference during treatment and before treatment comparing the rapamycin groups to the placebo group. I do not believe that is shown in this table. Could you more clearly show this difference?*

We thank the reviewer for this observation which has helped us to improve the data analysis. We now present a revised version of the repeated measures linear mixed model for ALSFRS-R, which include more coefficients than the previous one and which include explicit regression terms to test the differences in the change in the mean difference during treatment and before treatment comparing the rapamycin groups to the placebo group. We added the required data in Table 4, now showing also this between-groups analysis, comparing treatment arms with placebo (difference in monthly variation compared to before treatment period). We changed table 4 legend (page 24-25 of the main file, TC version) and statistical methods accordingly (page 15 of the main file, TC version).

9) *Line 336: It is not clear why this is considered a trend toward a slower decline given the width of the confidence interval.*

We apologize for the typo, erroneously remained after the last revision of the manuscript. We stopped the sentence concluding that there were no significant differences in the mean absolute change from baseline to each time point in the FVC and reporting mean differences at week 18 (page 7 lines 343-346 of the main file, TC version).

Reviewer #2 (Remarks to the Author):

1) *The authors responded to previous reviews. The phase 2 trial did not reach its primary outcome measure of increasing t reg number.*

We acknowledge the negative result of the trial both in the abstract and in the text. We further remarked the study limitations in the discussion (lines 512-518 page 11 of the main file, TC version): “The choice

of the primary outcome measure, as a binary response, has to be acknowledged as the main drawback of this study, together with the fact that Treg function, rather than their number would have been relevant for treatment effect. Indeed, we did not plan to study Treg suppressive function instead of their number alone and we were not able to assess Treg suppressor function as a post-hoc analysis due to lack of available samples. Before further studies in ALS, assessment of Treg suppressor function in response to different doses of rapamycin would be critical”.

- 2) *The authors were not able to assess t reg suppressor function.*

We further remarked this issue among limitations of the study on page 10, lines 486-488 of the main file, TC version (“the absence of data on Treg function requires further studies to ascertain what dose has the more beneficial immunological effect in ALS”), and on page 11, lines 514-517 of the main file, TC version (“Indeed, we did not plan to study Treg suppressive function instead of their number alone and we were not able to assess Treg suppressor function as a post-hoc analysis due to lack of available samples”).

- 3) *it is not clear that the conclusions of finding best dose hold.*

The lower dose resulted more stable in plasma and with less frequent AEs, but we agree with the reviewer that we don’t know what are the effects of different doses on Treg cells function. We removed from the abstract the conclusion on the best dose. We also reformulated considerations on the best dosage in the discussion, mainly centered on plasma stability and safety (page 10, lines 477-493 of the main file, TC version): “Besides demonstrating that rapamycin is safe in patients with ALS, we found that rapamycin dosage of 1 mg/m²/day ensured a better stability of plasma dosages (never overcoming toxicity threshold) that only seldom required dosages adjustment due to safety concerns. If the effect on Treg cells and other immunological outcome measures, was more evident for patients treated with rapamycin 1 mg/m²/day, the absence of data on Treg function requires further studies to ascertain what dose has the more beneficial immunological effect in ALS.”

- 4) *Before further studies in ALS, knowing whether t reg suppressor function is improved would be critical.*

We added this further consideration on page 11, line 517-518 of the main file, TC version: “Before further studies in ALS, assessment of Treg suppressor function in response to different doses of rapamycin would be critical”.

- 5) *While an explanation is provided for choosing 30% increase in t reg number, this isn't clear why the number would be relevant- rather than function.*

We clearly state this in the discussion, page 11, lines 512-514 of the main file, TC version: “The choice of the primary outcome measure, as a binary response, has to be acknowledged as the main drawback of this study, together with the fact that Treg function, rather than their number would have been relevant for treatment effect”.

- 6) *It may be that rapamycin should not be pursued further in als - or that there is need for another phase 2a trial to look at dose and other biomarkers (t reg suppression, etc).*

We remarked that “further studies should be performed to look at other biomarkers, such as Treg suppressor function and the best dose to establish if rapamycin deserves to be pursued in larger ALS trials” on page 9, lines 441-443 of the main file, TC version.

REVIEWER COMMENTS

Reviewer #1:

The authors have addressed my concerns. Please check the Pearson's correlation coefficients in lines 324-325 because the estimate is not within the confidence interval in either case so there must be an error.

RESPONSE TO REVIEWERS

Reviewer #1:

Reviewer #1 (Remarks to the Author):

"The authors have addressed my concerns. Please check the Pearson's correlation coefficients in lines 324-325 because the estimate is not within the confidence interval in either case so there must be an error."

We apologize for the error in the sign of the 95%CI. The right sentence is:

"Pearson's r coefficient was 0.18 (95% CI -0.50 to 0.17, p=0.308) between ALSFRS-R and serum NFL change (w18-w0), and 0.32 (95% CI -0.65 to 0.10, p=0.131) between ALSFRS-R and CSF NFL changes (w18-w0)."

We corrected the text accordingly.